# Emotional Intelligence as Critical Competence in Nurses’ Work Performance: A Cross-Sectional Study

**DOI:** 10.3390/healthcare12191936

**Published:** 2024-09-27

**Authors:** Petros Galanis, Aglaia Katsiroumpa, Ioannis Moisoglou, Konstantina Derizioti, Parisis Gallos, Maria Kalogeropoulou, Vasiliki Papanikolaou

**Affiliations:** 1Clinical Epidemiology Laboratory, Faculty of Nursing, National and Kapodistrian University of Athens, 11527 Athens, Greece; aglaiakat@nurs.uoa.gr (A.K.); parisgallos@nurs.uoa.gr (P.G.); mariakalo@nurs.uoa.gr (M.K.); 2Faculty of Nursing, University of Thessaly, 41336 Larisa, Greece; iomoysoglou@uth.gr; 3Faculty of Public Health Policy, University of West Attica, 11521 Athens, Greece; mlead21042@uniwa.gr (K.D.); bpapanikolaou@uniwa.gr (V.P.)

**Keywords:** emotional intelligence, work performance, job performance, nurses, emotions

## Abstract

Background/Objectives: Emotional intelligence may help nurses to cope with demanding work environments where the need to improve the quality and safety of the care provided, as well as the care of the chronically ill, prevails. Although it is well known that emotional intelligence is positively related to work performance, the literature on nurses is limited. The aim of our study was to examine the impact of emotional intelligence on work performance in a sample of nurses in Greece. Methods: We conducted a cross-sectional study with 318 nurses. We collected data from a convenience sample of nurses during January 2024. Since we conducted an online survey through social media, our sample could not be representative of all nurses in Greece. For instance, older nurses may be underrepresented in our study due to limited access on social media. We measured emotional intelligence with the Trait Emotional Intelligence Questionnaire-Short Form (TEIQue-SF) and work performance with the Individual Work Performance Questionnaire (IWPQ). We created multivariable linear regression models adjusted for sex, age, educational level, and work experience. We followed the Strengthening the Reporting of Observational Studies in Epidemiology (STROBE) guidelines. Results: We found that the four streams of emotional intelligence (i.e., well-being, self-control, emotionality, sociability) increased nurses’ work performance. In particular, we found a positive relationship between well-being and task performance (adjusted beta = 0.210, 95% CI = 0.140 to 0.281, *p*-value < 0.001) and contextual performance (adjusted beta = 0.135, 95% CI = 0.050 to 0.221, *p*-value = 0.002). Similarly, there was a positive relationship between self-control and task performance (adjusted beta = 0.136, 95% CI = 0.030 to 0.241, *p*-value = 0.012). Additionally, sociability increased task performance (adjusted beta = 0.223, 95% CI = 0.151 to 0.295, *p*-value < 0.001) and contextual performance (adjusted beta = 0.198, 95% CI = 0.111 to 0.286, *p*-value < 0.001). Moreover, emotionality (adjusted beta = −0.198, 95% CI = −0.319 to −0.076, *p*-value = 0.002) and sociability (adjusted beta = −0.133, 95% CI = −0.221 to −0.044, *p*-value = 0.003) reduced counterproductive work behavior. Conclusions: Our multivariable models identified a positive impact of emotional intelligence on nurses’ work performance. Nurse managers and healthcare organizations should adopt appropriate interventions to improve nurses’ emotional intelligence. Enhancing emotional intelligence among nurses can improve work performance and, thus, healthcare outcomes. Moreover, higher levels of emotional intelligence may empower nurses’ compassion and resilience, fostering a supportive work environment. In this context, the well-being of both nurses and patients may improve.

## 1. Introduction

Emotional intelligence is a multidimensional concept, and thus, a variety of definitions have been proposed by scholars over the years [1,2]. In short, there are two emotional intelligence models, namely, the trait model and the ability model. Trait emotional intelligence refers to personality traits that reflect individuals’ subjective emotional dispositions and experiences, including their ability to perceive, evaluate, and manage emotions effectively [3]. According to the ability model, emotional intelligence is a specific set of abilities focused on recognizing and managing emotions [4]. The literature suggests the positive effect of emotional intelligence on nurses’ psychosomatic health. In particular, higher emotional intelligence is associated with decreased risk for somatic complaints and lower burnout among nurses [5,6,7]. Moreover, studies including nurses have indicated the positive impact of emotional intelligence on work-related variables such as work engagement and mobbing [8,9,10,11].

In general, work performance refers to actions that manage problems within the working environment or individuals, such as self-control and problem-solving [12]. In particular, nurses’ work performance can be separated into two distinct categories: contextual performance and task performance [13,14]. Contextual performance in nurses can be considered as their effectiveness in carrying out tasks related to patient care. Additionally, task performance refers to nurses’ tasks regarding their institutional, social or/and psychological environment. Recently, scholars suggested a third dimension of work performance, namely counterproductive performance [15]. Counterproductive performance refers to a work behavior that has a detrimental impact on the organization’s well-being [16]. The literature suggests that several factors may affect nurses work performance, such as poor supervisor support, excessive stress, high levels of emotional labor, poor communication, lack of skills and training, and high levels of job strain [14,17,18,19]. On the other hand, scholars found positive relationships between psychological resilience, manager support, high levels of professional values, and collaborative leadership and work performance in nurses [20,21,22,23].

The positive relationship between emotional intelligence and job performance is well-established in the literature. In particular, three recent meta-analyses found that emotional intelligence positively correlates with job performance [1,2,24]. These meta-analyses used different approaches to examine the impact of emotional intelligence on job performance. In general, they classified studies into three categories: (a) studies that used self-report or peer-report measures, (b) studies that used “mixed models”, including trait models and ability models, and (c) studies that used ability-based models with objective test items. In short, meta-analyses found adjusted correlation coefficients between the streams of emotional intelligence and job performance between 0.24 and 0.30. Additionally, six studies until now have investigated the relationship between emotional intelligence and job performance in nurses [25,26,27,28,29,30,31]. Findings from studies in nurses are in accordance with the above meta-analyses indicating a positive relationship between emotional intelligence and job performance in nurses as well.

The management of healthcare organizations worldwide is facing major challenges regarding the quality and safety of health services, particularly in relation to patient care, as well as the care of the chronically ill. There is a positive correlation between the level of emotional intelligence in nurses and the quality and safety of nursing care [32,33]. Also, emotional intelligence is a crucial skill for nurses, as it enables them to experience less work-related stress while focusing on problem solving when caring for oncology patients [34]. Improved quality of care, increased patient satisfaction with the care provided, better patient-centered care and more compassionate attitudes towards patients with HIV are the benefits resulting from the impact of nurses’ emotional intelligence during the provision of care [35]. As emotional intelligence is a skill that can be acquired through training programs [36,37], the integration of such programs both at the undergraduate level and in the context of continuing professional development is of fundamental importance in the effort to improve the quality of care.

Since poor work performance has an impact on the quality of service delivery and nurses’ growth, health and well-being [23,38,39], it is essential to investigate the factors that may affect nurses’ work performance. As mentioned above, six studies until now have investigated the relationship between emotional intelligence and work performance among nurses [25,26,27,28,29,30,31]. However, these studies were conducted in four Asian countries, i.e., India, Jordan, Saudi Arabia and Iran. Since there is a dearth of similar studies in European countries, we aimed to examine the impact of emotional intelligence on work performance in a sample of nurses in a European country. To our knowledge, this was the first study that was conducted in a European country. Moreover, the identification of predictors of nurses’ work performance is crucial to improving productivity and the quality of healthcare. Thus, in this context, our study aimed to investigate whether emotional intelligence can predict nurses’ work performance.

## 2. Materials and Methods

### 2.1. Study Design

We conducted a cross-sectional study in Greece. We collected data from nurses during January 2024. We conducted an online survey approaching nurses through nurse groups on Facebook, Instagram, and LinkedIn. Thus, our study questionnaire was available only for nurses since we posted it only in nurse groups. These nurse groups were created by nurse administrators who check the professional status of participants. For instance, administrators of nurse groups ask potential members to provide the name of the hospital at which they work, and the hospital ward they work on. After confirmation of their professional status, administrators provide access to their nurse groups. We developed an online version of our study questionnaire using Google forms. Then, we posted the study questionnaire on social media, inviting nurses to participate in our study. Thus, we obtained a convenience sample. Our inclusion criteria included the following information: (a) nurses who have been working in clinical settings, (b) with clinical experience of at least two years, and (c) Greek-speaking nurses. We followed the Strengthening the Reporting of Observational Studies in Epidemiology (STROBE) guidelines [40].

Considering a low effect size (f^2^ = 0.05) of emotional intelligence on work performance, the number of independent variables (four predictors and four confounders), a confidence level of 95%, and a margin of error of 5%, the sample size was estimated at 262 nurses.

### 2.2. Measures

We measured the following demographic and job characteristics: sex (females or males), age (continuous variable), educational level (MSc/PhD diploma or not), and work experience (continuous variable).

We measured emotional intelligence with the Trait Emotional Intelligence Questionnaire-Short Form (TEIQue-SF) [41]. The TEIQue-SF contains 30 items, and answers are on a 7-point Likert scale from 1 (completely disagree) to 7 (completely agree). The TEIQue-SF includes four factors, i.e., well-being, self-control, emotionality, and sociability. We chose TEIQue-SF to measure emotional intelligence due to the high reliability and validity of the scale. In particular, a recent meta-analysis with 426 studies found Cronbach’s alpha for total TEIQue-SF, well-being, self-control, emotionality, and sociability as 0.86, 0.79, 0.68, 0.70, and 0.69, respectively [42]. Moreover, two systematic reviews supported the validity of the TEIQue-SF [43,44]. The score on each factor ranged from 1 (low levels of emotional intelligence) to 7 (high levels of emotional intelligence). We used the valid Greek TEIQue-SF [45]. In our study, Cronbach’s alpha for the TEIQue-SF, well-being, self-control, emotionality, and sociability was 0.851, 0.816, 0.742, 0.888, and 0.767, respectively.

We measured nurses’ work performance with the Individual Work Performance Questionnaire (IWPQ) [46]. The IWPQ includes 18 items with a recall period of three months and contains three factors, i.e., task performance, contextual performance, and counterproductive work behavior. An extended, systematic review found that the most frequent dimensions that are used to describe individual work performance are task performance, contextual performance, and counterproductive work behavior [15]. Since the IWPQ measures these three dimensions comprehensively and generically, we chose to use this scale in our study [46]. The IWPQ is a general tool that can be used for any work, and thus, we did not make any adaptations for the nursing profession. Answers are on a 5-point Likert scale from 0 (seldom for task and contextual performance, never for counterproductive work behavior) to 4 (always for task and contextual performance, often for counterproductive work behavior). The score on each factor ranges from 0 to 4. Higher scores indicate higher task performance, contextual performance, and counterproductive work behavior. We used the valid Greek IWPQ [47]. In our study, Cronbach’s alpha for the IWPQ, task performance, contextual performance, and counterproductive work behavior was 0.862, 0.741, 0.885, and 0.775, respectively.

### 2.3. Ethical Issues

The Ethics Committee of the Faculty of Nursing, National and Kapodistrian University of Athens approved our study protocol (approval number: 464, approval date: October 2023). We conducted our study on an anonymous and voluntary basis following the Declaration of Helsinki [48]. An information sheet on the online questionnaire informed nurses about the aim and the design of our study. Afterward, we asked nurses if they wanted to participate in our study by completing the questionnaire. Nurses with a positive answer could further proceed with the study questionnaire. Thus, we obtained informed consent from nurses participating in our study.

After data collection, we downloaded them through Google Forms as an Excel file. We encrypted the Excel file with a password known only to the researchers who conducted the study. Afterward, we transferred data in a SPSS file that was also encrypted with a password.

### 2.4. Statistical Analysis

We use numbers and percentages to present categorical variables. Moreover, we use the mean, standard deviations (SD), median, and range to present continuous variables. We used the Kolmogorov–Smirnov test and Q-Q plots to examine the distribution of continuous variables. We found that scores on task performance, contextual performance, and counterproductive work behavior followed normal distribution. Thus, we used univariate and multivariable linear regression analysis to examine the impact of emotional intelligence on work performance. We considered demographic and job characteristics (i.e., sex, age, educational level, and work experience) as potential confounders in the relationship between emotional intelligence and work performance. We chose sex, age, educational level, and work experience as potential confounders since several studies identified these variables as significant factors in determining nurses’ performance [21,49,50,51,52,53]. We finally constructed multivariable linear models by eliminating the confounding caused by demographic and job characteristics. Since our analysis included three dependent variables (i.e., task performance, contextual performance, and counterproductive work behavior) and four independent variables (i.e., well-being, self-control, emotionality, and sociability), we performed multivariate multiple linear regression analysis. Our omnibus null hypothesis was that all regression coefficients are equal to zero across all dependent variables. We used Wilk’s lambda to test the omnibus hypothesis [54]. We checked assumptions for the multivariate multiple linear regression analysis, such as the linear relationship between independent and dependent variables, the absence of outliers, the presence of homoscedasticity, the normality of residuals, and absence of multicollinearity. We present unadjusted and adjusted unstandardized coefficients beta, 95% confidence intervals (CIs), adjusted standardized coefficients beta, *p*-values, and R^2^. We considered *p*-values < 0.05 as statistically significant. We used IBM SPSS Statistics for Windows, Version 21.0. Armonk, NY, USA: IBM Corp.) for statistical analysis.

## 3. Results

### 3.1. Participant Characteristics

The study population included 318 nurses. Table 1 presents the demographic and job characteristics of nurses. The mean age of nurses was 41.2 years (SD; 11.2), with a median value of 41 years and a range from 24 to 65 years. Among our nurses, 56.9% were females, and 23.3% possessed a MSc/PhD diploma. The mean number of years of work experience was 14.9 (SD; 10.2, median; 11, range; 2 to 40).

### 3.2. Study Scales

Table 2 shows the descriptive statistics for our study scales. Mean scores on the TEIQue-SF and sub-factors indicated moderate to high levels of emotional intelligence in our sample. Regarding emotional intelligence, we found higher emotionality levels, followed by well-being, self-control, and sociability levels.

Our nurses showed moderate levels of task performance and contextual performance and low levels of counterproductive work behavior.

### 3.3. Impact of Emotional Intelligence on Work Performance

Our multivariate multiple analysis identified a positive impact of emotional intelligence on nurses’ work performance (Table 3). In particular, Wilk’s lambda was statistically significant (F [3, 306] = 21.951, *p*-value < 0.001). Therefore, one or more of our four independent variables (i.e., well-being, self-control, emotionality, and sociability) were statistically significant predictors of one or more of our three dependent variables (i.e., task performance, contextual performance, and counterproductive work behavior). For the model with task performance as the dependent variable, R^2^ = 36.8%, F = 22.317, and *p*-value < 0.001. For the model with contextual performance as the dependent variable, R^2^ = 26.6%, F = 14.224, and *p*-value < 0.001. For the model with counterproductive work behavior as the dependent variable, R^2^ = 13.3%, F = 6.179, and *p*-value < 0.001.

In particular, we found a positive relationship between well-being and task performance (adjusted beta = 0.210, 95% CI = 0.140 to 0.281, *p*-value < 0.001) and contextual performance (adjusted beta = 0.135, 95% CI = 0.050 to 0.221, *p*-value = 0.002). Similarly, there was a positive relationship between self-control and task performance (adjusted beta = 0.136, 95% CI = 0.030 to 0.241, *p*-value = 0.012) and contextual performance (adjusted beta = 0.295, 95% CI = 0.167 to 0.423, *p*-value < 0.001). Additionally, sociability increased task performance (adjusted beta = 0.223, 95% CI = 0.151 to 0.295, *p*-value < 0.001) and contextual performance (adjusted beta = 0.198, 95% CI = 0.111 to 0.286, *p*-value < 0.001). Moreover, we found a negative relationship between emotionality (adjusted beta = −0.198, 95% CI = −0.319 to −0.076, *p*-value = 0.002) and sociability (adjusted beta = −0.133, 95% CI = −0.221 to −0.044, *p*-value = 0.003) with counterproductive work behavior.

## 4. Discussion

To the best of our knowledge, this study is the first to investigate the impact of emotional intelligence on work performance among nurses in Europe. After adjusting for several confounders, we found that emotional intelligence improved nurses’ work performance. In particular, emotional intelligence increases task performance and contextual performance and decreased counterproductive work behavior. Moreover, our results suggest moderate levels of task performance and contextual performance, low levels of counterproductive work behavior, and moderate to high levels of emotional intelligence.

Considering the findings of our multivariate multiple linear regression analysis, our four independent variables (i.e., well-being, self-control, emotionality, and sociability) were better predictors (i.e., explained more variance) of task performance than of contextual performance and counterproductive work behavior since R^2^ for the task performance was higher than R^2^ for the other two dependent variables. Therefore, our findings suggested that emotional intelligence has a greater impact on task performance than contextual performance and counterproductive work behavior. The fact that task performance refers to the proficiency with which workers perform tasks or the core substantive central to their work may explain our findings [55,56]. Traditionally, task performance has been the main focus of individual work performance and often includes behaviors such as job skills, work quality and quantity, and job knowledge [16,56]. Additionally, our standardized regression coefficients allowed for a comparison of the relative contribution of well-being, self-control, emotionality, and sociability on task performance, contextual performance, and counterproductive work behavior. Thus, for the dependent variable “task performance”, the variables “sociability” and “well-being” seemed to make the greatest relative contribution. Moreover, for the dependent variable “contextual performance”, the variables “sociability” and “self-control” seemed to make the greatest relative contribution. On the other hand, for the dependent variable “counterproductive work behavior”, the variables “sociability” and “emotionality” seemed to make equal contributions.

In our study, we found moderate levels of task and contextual performance and low levels of counterproductive work behavior among nurses. Three studies [57,58,59] used the Individual Work Performance Questionnaire to assess levels of work performance among nurses, and thus, we were able to make direct comparisons with our findings. In particular, Al-Dossary et al. in Saudi Arabia [57] and Jasiński et al. in Poland [59] found similar levels of task performance, and contextual performance with our nurses but higher levels of counterproductive work behavior. Moreover, levels of task performance, contextual performance and counterproductive work behavior among nurses in Turkey [58] were considerably higher than among nurses in our study.

Additionally, our nurses showed moderate-to-high levels of emotional intelligence. Several studies in Greece, the USA, and South Africa [60,61,62,63] also used the Trait Emotional Intelligence Questionnaire-Short Form to measure emotional intelligence in nurses. These studies support our findings since the mean TEIQue-SF score in our study was 5.56, while in the other studies, it ranged from 4.89 to 5.42.

The main finding of our study is the positive relationship between emotional intelligence and work performance in nurses. After adjustment for several confounders, we found that emotional intelligence affects the three dimensions of work performance. In particular, our multivariable regression models identified that nurses with higher levels of emotion show higher levels of task performance and contextual performance and lower levels of counterproductive work behavior. Emotional intelligence contributes to improved job performance among nurses by decreasing occupational stress. High emotional intelligence enables individuals to exhibit better emotional behaviors in their work environment and experience less stress, leading to increased productivity and enhanced performance [64,65,66]. Emotional intelligence is founded on the capacity to perceive, regulate, and comprehend emotions, all of which are crucial for achieving high job performance [24]. Individuals with healthy emotion regulation skills can choose more effective approaches to job demands, which allows them to conserve resources and maintain high levels of job performance [24,67]. Employees with high emotional intelligence tend to be more optimistic and possess the ability to manage and change stressful situations in their work environment without being influenced by external factors [68]. Therefore, these employees also actively respond to occupational stress. Several emotional intelligence components and outcomes are applicable to the workplace, including both ability-based and trait-based aspects, which lead to greater adaptability and, ultimately, better job performance [2]. Emotional intelligence is the capacity to utilize emotions to enhance reasoning. Consequently, emotionally intelligent employees should be able to gain more knowledge about tasks from emotion-related situations [2]. The ability to recognize emotions in oneself and others is vital for effective social interactions, as is the ability to regulate one’s own emotions [1]. The ability to recognize emotions in others may help individuals determine when to perform emotional labor, just as the ability to recognize one’s own emotions may help employees recognize when they need to adjust their emotional expressions [1].

Several studies in India, Saudi Arabia, Iran and Jordan [25,26,27,28,29,30,31] confirmed the positive relationship between emotional intelligence and work performance in nurses. All studies except one that was conducted by Vahidi et al. [31] used multivariable regression models to eliminate confounders as we did in our study. Two studies [26,29] used the Individual Work Performance Questionnaire to measure nurses’ work performance in a similar way to our study. Other valid scales such as the Six-Dimension Scale of Nursing Performance were used to assess work performance. Moreover, all studies used different valid tools to measure nurses’ emotional intelligence than our tool, i.e., the Trait Emotional Intelligence Questionnaire-Short Form. In particular, scholars used the Genos Emotional Intelligence Assessment, the Wong & Law Emotional Intelligence scale, the Schutte Emotional Intelligence Scale, and the Bar-On Questionnaire.

Several recent studies in Greece revealed high levels of quiet quitting, job burnout, job dissatisfaction, workplace mobbing, and turnover intention among nurses [69,70,71,72,73,74]. Therefore, nurses in Greece have to face many challenges, which threaten their work performance and, thus, the quality of nursing care. In this context, high levels of emotional intelligence among nurses are critical to improving the quality of nursing care in Greece. A recent review identified the positive association between emotional intelligence and both physical and emotional caring [35]. Similarly, a study in Greece revealed a positive correlation between emotional intelligence and the frequency of occurrence and the importance of caring behaviors in a sample of mental health nurses [75].

Our study has several limitations. First, we obtained a convenience sample of nurses through social media. Thus, we cannot generalize our findings. Further studies with random and representative samples should be conducted. Second, we used valid scales to measure emotional intelligence and work performance, but information bias is probable in our study due to the self-reported design of these scales. Third, the cross-sectional design of our study prevented us from establishing a causal relationship between emotional intelligence and work performance. Fourth, our sample included nurses from a European country with specific clinical work environments. Thus, further studies should be conducted in other European countries to extract more valid results. Finally, we eliminated confounding caused by several demographic and job variables. However, several other variables can act as confounders in the relationship between emotional intelligence and work performance. Thus, scholars should eliminate more confounders in future research to obtain more valid results.

## 5. Conclusions

Our results suggest the positive impact of emotional intelligence on work performance among nurses. Thus, emotional intelligence may be especially important in healthcare services since nurses interact very often with individuals. Considering the limited literature in this field on the population of nurses and the limitations of our study, further research should be conducted to infer more valid conclusions. Special attention should be given by scholars in Europe, Northern America, and Australia since there are no studies on the relationship between emotional intelligence and work performance. Further studies, including random and stratified samples from the population of nurses, may add significant evidence to this field. Similarly, studies in different countries and clinical settings may improve our knowledge regarding the relationship between emotional intelligence and work performance. Scholars may also investigate the potential mediating role of several work-related variables, such as job dissatisfaction, job burnout, quiet quitting, and coping strategies, in the relationship between emotional intelligence and work performance. Nurse managers, healthcare policymakers, and healthcare organizations should develop and adopt appropriate interventions to improve emotional intelligence skills in nurses. Enhancing emotional intelligence among nurses can improve work performance and, thus, healthcare outcomes. Moreover, higher levels of emotional intelligence may empower nurses’ compassion and resilience, fostering a supportive work environment. In this context, the well-being of both nurses and patients may improve. The development of emotional intelligence abilities among nurses facilitates successful change processes in healthcare organizations.

## Figures and Tables

**Table 1 healthcare-12-01936-t001:** Demographic and job characteristics of nurses (N = 318).

Characteristics	N	%
Sex		
Males	137	43.1
Females	181	56.9
Educational level		
University degree	244	76.7
MSc/PhD diploma	74	23.3

**Table 2 healthcare-12-01936-t002:** Descriptive statistics for the study scales (N = 318).

Scale	Mean	Standard Deviation	Median	Range (Min–Max)
Emotional intelligence	5.56	0.47	5.57	2.37 (4.27–6.63)
Well-being	5.44	0.96	5.67	5.00 (2.00–7.00)
Self-control	5.48	0.65	5.33	2.67 (4.33–7.00)
Emotionality	5.99	0.68	6.00	2.75 (4.25–7.00)
Sociability	4.85	0.95	5.00	4.83 (2.00–6.83)
Work performance				
Task performance	2.51	0.67	2.60	3.40 (0.60–3.40)
Contextual performance	2.42	0.76	2.50	3.50 (0.50–4.00)
Counterproductive work behavior	1.34	0.70	1.40	3.40 (0.00–3.40)

**Table 3 healthcare-12-01936-t003:** Multivariate multiple regression analysis with nurses’ work performance as the dependent variable (N = 318).

		Task Performance							Contextual Performance							Counterproductive Work Behavior						
	Univariate Models				Multivariable Model ^a,b^			Univariate Models				Multivariable Model ^a,c^			Univariate Models					Multivariable Model ^a,d^		
	Unadjusted Coefficient Beta	95% CI for Beta	*p*-Value	Adjusted Unstandardized Coefficient Beta	95% CI for Beta	Adjusted Standardized Coefficient Beta	*p*-Value	Unadjusted Coefficient Beta	95% CI for Beta	*p*-Value	Adjusted Unstandardized Coefficient Beta	95% CI for Beta	Adjusted Standardized Coefficient Beta	*p*-Value	Unadjusted Coefficient Beta		95% CI for Beta	*p*-Value	Adjusted Unstandardized Coefficient Beta	95% CI for Beta	Adjusted Standardized Coefficient Beta	*p*-Value
Well-being	0.323	0.254 to 0.392	<0.001	0.210	0.140 to 0.281	0.292	<0.001	0.268	0.186 to 0.350	<0.001	0.135	0.050 to 0.221	0.161	0.002	−0.147		−0.226 to −0.067	<0.001	−0.056	−0.142 to 0.031	−0.076	0.206
Self-control	0.328	0.220 to 0.437	<0.001	0.136	0.030 to 0.241	0.129	0.012	0.445	0.325 to 0.564	<0.001	0.295	0.167 to 0.423	0.250	<0.001	−0.150		−0.269 to −0.031	0.013	0.011	−0.119 to 0.140	0.010	0.873
Emotionality	0.207	0.100 to 0.314	<0.001	−0.013	−0.112 to 0.086	−0.014	0.790	0.241	0.121 to 0.362	<0.001	−0.012	−0.132 to 0.109	−0.011	0.851	−0.241		−0.353 to −0.130	<0.001	−0.198	−0.319 to −0.076	−0.191	0.002
Sociability	0.356	0.288 to 0.424	<0.001	0.223	0.151 to 0.295	0.341	<0.001	0.331	0.250 to 0.411	<0.001	0.198	0.111 to 0.286	0.277	<0.001	−0.186		−0.265 to −0.106	<0.001	−0.133	−0.221 to −0.044	−0.177	0.003

^a^ Multivariable models are adjusted for sex, age, educational level, and work experience; ^b^ R^2^ for the multivariable model = 36.8%, *p*-value for ANOVA < 0.001; ^c^ R^2^ for the multivariable model = 26.6%, *p*-value for ANOVA < 0.001; ^d^ R^2^ for the multivariable model = 13.3%, *p*-value for ANOVA < 0.001; and CI: confidence interval.

## Data Availability

The data presented in this study are available upon request from the corresponding author.

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
