# Peer review of "Emotional Intelligence as Critical Competence in Nurses’ Work Performance: A Cross-Sectional Study"

_healthcare, 2024, doi:10.3390/healthcare12191936_

Round 1

Reviewer 1 Report

Comments and Suggestions for Authors

Esteemed authors, 

Thank you for allowing me to review this article of significant importance which focuses on Emotional intelligence as a critical competence in nurses' work performance.

The title is clear and coherent with the aim and content of the article, although putting ‘evidence from Greece’ in the title may mislead readers into considering that this study may be representative of the population or the data generalizable, which is not the case. I leave this for the authors to consider. 

Abstract

This begins with an introductory section on the topic, stating the aim of the study and its methodology. However, the authors could mention the sample size in the abstract, giving an overview of the scope of the study and anticipating how representative the sample may be. 

Line 23-25 the strength of the findings can be reinforced with the statistical values obtained, p-values or confidence intervals. 

Introduction

The fundamental concepts of the study are well defined and similar studies carried out in other countries have been mobilised. It is not clear whether there is any other reason for the relevance of the study in Greece apart from the ‘dearth of similar studies’ line 92. 

Methods 

Was the questionnaire available for all social media users to access? How was it ensured that the responses obtained were from nurses?

It is not clear how consent was obtained, only that it was given line 136-137. How was the purpose of the study explained to the participants so that they could give their informed consent? The information is missing and its description could provide transparency on the ethical issues of the study. 

How was the data stored and protected? 

Results

The statistical analysis is detailed and the tables are easy to interpret. 

Discussion 

The findings are analyzed and compared with the available evidence on the subject. It may become clearer how useful the findings are in clinical practice, namely in the quality of nursing care in Greece. The limitations of the study are recognized and detailed. 

Conclusions 

It could provide more concrete suggestions for further study on this topic, aimed at nursing research and practice. 

References

The references used are current and relevant to the type of study presented. 

I would like to congratulate the authors on their work; however, considering the various suggestions provided, I believe the article requires a major review. 

Author Response

Dear Reviewer, 

Reviewer 2 Report

Comments and Suggestions for Authors

Thank you for the opportunity to read this manuscript about the emotional intelligence of nurses and work environments. 

Abstract

Why have you specified the aim of the sample and the country? This consideration is also valid for the title. Is this concept and research influenced by local conditions? 

The number of participants, the equator checklist used and the implications of this research are missing in the abstract.

Introduction

The introduction is well written and there are explanations of the included variables. Consider focusing on the Greek condition or remaining general without specifying the country (declare only a European country) and revise the aim.

In the methodology, you should declare the STROBE checklist used for conducting and reporting this research. Add it as a supplementary file.

Remove age from Table 1, which reports only frequencies. 

Table 2 reports the range that typically is the min and maximum. In this case, what is the range?

Insert in the conclusion some implications for healthcare organizations and managers.

Author Response

Dear Reviewer, 

Reviewer 3 Report

Comments and Suggestions for Authors

This manuscript provides valuable insights into the impacts of emotional intelligence (EI) on nurses' work performance, but there are areas for improvement. 1) The rationale for selecting the TEIQue-SF to measure EI needs clarification—why was this specific tool chosen over other measures? 2) While the IWPQ is used to assess nurses' work performance, it is important to justify whether this tool is appropriate for this context or if adaptations are necessary for the nursing profession. 3) In the MLG model, it is unclear why variables such as sex, age, education, and work experience are controlled. Providing evidence from the literature would strengthen the argument for these choices.

Author Response

Dear Reviewer, 

Reviewer 4 Report

Comments and Suggestions for Authors

I think this is a significant study that confirms that nurses' emotional intelligence is a major influence on job performance. Let me share some thoughts on the results and discussion section.

1)    In addition to the normal distribution assumption, you must also show that the assumptions for multivariate linear regression are met.

2)    This study presents the results of a multivariate multiple regression analysis, and the description of the results is in the form of a multiple regression analysis. It would be good to explain the difference in interpreting the results of the two analyses.

3)    The results of a multivariate multiple regression analysis are more informative than a multiple regression analysis and can take into account inter-correlationships among the dependent variables. The current discussion technique appears to be a simple repetition of the study results. The discussion needs to be described more comprehensively, taking into account the advantages of the interpretation of multivariate multiple regression analysis.

That's all for now.

Yours sincerely, Reviewer

Author Response

Dear Reviewer, 

Round 2

Reviewer 1 Report

Comments and Suggestions for Authors

Dear authors, 

After making the necessary corrections, I believe the article is now more coherent and provides readers with new evidence on the subject being studied. I appreciate the opportunity to review your work and help clarify certain aspects.

I hope your research reaches great success in reaching a wide audience.